# RadioLigand Therapy with [^177^Lu]Lu-PSMA-617 for Salivary Gland Cancers: Literature Review and First Compassionate Use in France

**DOI:** 10.3390/ph16050754

**Published:** 2023-05-16

**Authors:** Marie Terroir, Chloé Lamesa, Mehdi Krim, Lavinia Vija, Jean-Sébastien Texier, Thibaut Cassou-Mounat, Jean-Pierre Delord, Delphine Vallot, Frédéric Courbon

**Affiliations:** 1Nuclear Medicine Department, Institut Claudius Regaud, Institut Universitaire du Cancer de Toulouse, CEDEX 09, 31059 Toulouse, France; 2Radiopharmacy Department, Institut Claudius Regaud, Institut Universitaire du Cancer de Toulouse, CEDEX 09, 31059 Toulouse, France; 3Medical Oncology Department, Institut Claudius Regaud, Institut Universitaire du Cancer de Toulouse, CEDEX 09, 31059 Toulouse, France; 4Physics Department, Institut Claudius Regaud, Institut Universitaire du Cancer de Toulouse, CEDEX 09, 31059 Toulouse, France

**Keywords:** salivary gland cancer, vectored radioligand therapy, [^177^Lu]Lu-PSMA-617

## Abstract

Salivary gland cancers are rare tumors comprising a large group of heterogeneous tumors with variable prognosis. Their therapeutic management at a metastatic stage is challenging due to the lack of therapeutic lines and the toxicity of treatments. [^177^Lu]Lu-PSMA-617 (prostate-specific membrane antigen) is a vectored radioligand therapy (RLT) initially developed to treat castration-resistant metastatic prostate cancer with encouraging results in terms of efficacy and toxicity. Many malignant cells could be treated with [^177^Lu]Lu-PSMA-617 as long as they express PSMA as a consequence of androgenic pathway activation. RLT may be used when anti-androgen hormonal treatment has failed, particularly in prostate cancer. [^177^Lu]Lu-PSMA-617 has been proposed in certain salivary gland cancers, though the expression of PSMA is demonstrated by a significant uptake using [^68^Ga]Ga-PSMA-11 PET scan. This theranostic approach could be a new therapeutic option, warranting prospective investigation in a larger cohort. We review the literature on this subject and offer a clinical illustration of compassionate use in France as a perspective for administering [^177^Lu]Lu-PSMA-617 in salivary gland cancer.

## 1. Introduction

Salivary gland cancers are rare tumors comprising a large group of heterogeneous tumors with many histopathological variants (24 types according to the WHO 2022) [1] and variable prognosis (Table 1).

The two most frequently encountered types are mucoepidermoid carcinoma and adenoid cystic carcinoma (ACC). The therapeutic options are essentially based on surgery, combined, whenever possible, with radiation therapy, then cytotoxic chemotherapy and, eventually, targeted therapies, depending on the mutation status of the tumor. These treatments are burdensome with variable therapeutic responses and adverse effects that are sometimes difficult to control. In the event of relapse or metastatic evolution, the therapeutics options are limited.

According to the recommendations issued in 2022 by the French Network of Rare Head and Neck Tumors “Réseau d’Expertise Français sur les Cancers ORL Rares” (REFCOR) [4], there is no standard of care for these tumors. In the event of loco-regional recurrence, several options are possible: revision surgery, if possible, followed by postoperative radiotherapy; exclusive radiotherapy if surgery is unfeasible or refused; uni- or bilateral lymph node dissection in the event of recurrence; and chemotherapy in case the tumor is high grade (no standard). Unfortunately, these local strategies are limited by the technical feasibility of surgery and the radiation doses already administered in these areas. At the metastatic stage, the therapeutic strategy depends on histopathologic findings.

Locoregional therapy is preferable for ACC. For patients with recurrent oligometastatic disease (one to five metastases), local treatment (surgery such as metastasectomy, stereotactic irradiation, or interventional radiology) can be considered, thereby delaying the introduction of systemic treatments. For patients with poly-metastatic ACC (>5 metastases), initial management is based first on watchful waiting to assess the rate of progression. This initial surveillance without treatment can sometimes last several months or even years [4]. Systemic treatment will be initiated only if the disease has evolved by more than 20% in the previous 6 months, if it is life threatening or impairs the functional prognosis, or if symptoms worsen the general condition of the patient. In these settings, the recommended first-line treatment is the combination of cisplatin + vinorelbine [5] or cisplatin + doxorubicin + cyclophosphamide [6,7].

For indolent metastatic non-ACC salivary gland tumors, the initial management may be based on an initial surveillance phase. The criteria for initiating treatment are the same as for ACC (progression > 20% in 6 months, symptomatic disease, threatening lesion). Some histological subtypes are more aggressive and require systemic treatment, particularly salivary ductal carcinomas (SDC) and most not otherwise specified (NOS) adenocarcinomas. If chemotherapy is chosen, the combination of carboplatin and taxane is recommended as first-line treatment for inoperable relapsed and/or metastatic non-ACC disease [5,8], or cisplatin + doxorubicin + cyclophosphamide as an alternative [6,7]. In salivary gland carcinomas (excluding ACC), it is recommended to look for HER2 [9], androgen receptors (AR) [10,11], NTRK fusion (for mammary analog secretory carcinoma (MASC) tumors) [12] overexpression by immunohistochemistry. It is also recommended to perform a fuller molecular biology analysis of the tumor, if possible, using next-generation sequencing (NGS). In the event of Her2 positivity, it is recommended to add trastuzumab (off-label use) to the first line of chemotherapy: carboplatin + taxane + trastuzumab. Regional or national consensus group agreement is required if cytotoxic chemotherapy ± anti-HER2 treatment, antiandrogenic treatment (complete androgenic blockade: anti-androgens and LHRH agonists) or targeted therapy (Pazopanib) is chosen. The decision depends on the tumor progression and on the patient’s symptoms, age and co-morbidities. If possible, access to clinical trials is generally recommended.

Chemotherapy, anti-HER2 treatment and targeted therapy present several toxicities such as HBP (high blood pressure) and aplasia, which are more or less easy to control. Based on the observation that some salivary gland carcinomas showed significant uptake in [^68^Ga]Ga-PSMA-11 PET-CT (Positron Emission Tomography—Computed Tomography) [13,14,15] and in view of the encouraging results of patients treated for castration-resistant prostate cancer [16], some teams treat metastatic salivary gland carcinoma by [^177^Lu]Lu-PSMA-617. Indeed, it has been observed that salivary gland carcinomas, in particular certain subtypes such as ACC and salivary duct carcinoma, show an expression of PSMA either on the tumor cells or in the neovascularization of these tumor [2,3]. The level of PSMA expression is variable but significant in ACC, and less important in salivary duct carcinoma.

## 2. Use of [^177^Lu]Lu-PSMA-617 in France

### 2.1. [^177^Lu]Lu-PSMA-617 Used in Castration-Resistant Prostate Cancer

[^177^Lu]Lu-PSMA-617 is a radioligand therapy (RLT) that has been used in recent years for castration-resistant prostate cancer. In France, early access authorization (EAA) allows its use in adults with progressive castration-resistant prostate cancer expressing PSMA receptors and previously treated with taxane chemotherapy (for eligible patients) and at least one so-called second-generation hormonal therapy. These eligibility criteria are based on the results of the international phase III VISION trial [16]. This trial evaluated the use of [^177^Lu]Lu-PSMA-617 in the treatment of metastatic castration-resistant prostate cancer (mCRPC); this study enrolled over 800 patients with mCRPC and compared treatment with [^177^Lu]Lu-PSMA-617 in combination with best supportive care to standard chemotherapy or hormone therapy. Patients were randomized to receive either [^177^Lu]Lu-PSMA-617 combined with the best supportive care or standard treatment. The results showed a statistically significant improvement in overall survival, progression-free survival and quality of life in patients treated with [^177^Lu]Lu-PSMA-617 in combination with the best supportive care compared to standard treatment. Regarding toxicities, there were more adverse events in the [^177^Lu]Lu-PSMA-617 arm. Fatigue, nausea and dry mouth were the most common adverse events, but only evaluated as grade 1 or 2 of CTCAE. Xerostomia is one of the common adverse effect reported with [^177^Lu]Lu-PSMA-617. Fortunately, some studies about [^177^Lu]Lu-PSMA-617 in prostate cancer showed that xerostomia more often remains at grade 1 or 2 [17,18]. In addition, in a VISION study, a [^68^Ga]Ga-PSMA-11 PET-CT is required to decide on the eligibility for treatment with [^177^Lu]Lu-PSMA-617 [19]. This verifies that the uptake of the tracer and therefore of the drug is sufficient to be effective. There is no well-established rule to select patients for treatment using imaging. The concept of patient selection by pre-therapeutic imaging allowing the “projection” of drug fixation during the therapeutic phase is called “theranostics”, i.e., using the same molecule for both PET diagnosis and patient eligibility for treatment by RLT. Theranostics is gaining more and more importance in nuclear medicine, particularly in oncology.

### 2.2. [^177^Lu]Lu-PSMA-617 Therapy Used in Salivary Gland Cancer

In recent years, a few publications with a small number of patients and clinical cases have shown that [^68^Ga]Ga-PSMA-11 PET-CT can show significant uptake in some tumors other than the prostate such as those of salivary origin (ACC and other salivary carcinomas that can include up to 20 different histological types—Table 1) [13,20,21]. Thus, in ACC, the intensity of the uptake was variable with a mean Standard Uptake Value maximum (SUVmax, which is the tool to measure the intensity of uptake) ranging from 2.41 to 13.8 in cases of local recurrence and from 2.04 to 14.9 for distant metastases. For other salivary gland carcinomas, the mean SUVmax ranged from 1.2 to 12.5. A few teams have proposed treatment by RLT with [^177^Lu]Lu-PSMA-617 to patients who no longer have any conventional therapeutic options [14,22]. For this purpose, all the relevant studies containing information on [^177^Lu]Lu-PSMA-617 therapy in salivary gland cancer were included from Pubmed between 2018 and March 2023. We obtained three relevant publications: one case report [22], one retrospective study about six patients [23] and one retrospective study combining [^68^Ga]Ga-PSMA-11 PET CT and [^177^Lu]Lu-PSMA-617 therapy [24] (Table 2).

One publication showed good tolerance and the possible use of [^177^Lu]Lu-PSMA-617 in salivary gland carcinomas for which conventional treatment was no longer possible [23]. This retrospective study included patients with increasing tumor-related discomfort, either during active follow-up or referred for a second opinion, deemed irresectable and without other standard palliative treatment options. The eligibility of patients for [^177^Lu]Lu-PSMA-617 treatment was based on the outcome of two ways: analyzing the tumor tissue PSMA expression by immunohistochemistry and on the tumor uptake above the liver SUVmax on [^68^Ga]Ga-PSMA-11 PET-CT. In each case, the local multidisciplinary tumor board validated the treatment by [^177^Lu]Lu-PSMA-617. Patients were treated in a compassionate use program with four courses of 6 to 7.4 GBq of [^177^Lu]Lu-PSMA-617 every 6 to 8 weeks, as commonly used for prostate treatment. Response to treatment was assessed by questionnaire and [^68^Ga]Ga-PSMA-11 PET-CT every two cycles. The [^68^Ga]Ga-PSMA-11 PET-CT assessment used a mix of RECIST and PERCIST type evaluation applied to this tracer [25,26]. In fact, in this study, the authors considered a progression disease with a cut-off of more than 30% of the SUVmax in the hottest lesion and a decrease of more than 20% of the SUVmax in the hottest lesion for partial response between the pre- and post-treatment imaging. Between the two situations, they considered the disease as stable, and a complete response corresponded to the disappearance of all tumoral uptakes. In case of clinical or radiological progression, the RLT was stopped. Six patients were treated. Two patients had a response to treatment or disease considered stable for approximately 6 and 10 months, while the disease progressed rapidly in the other four patients. Treatment toxicities were mainly reported as grade 1 and 2; only one case of grade 3 was observed with thrombocytopenia [27]. There was no dosimetric study in this article. This admittedly modest study nevertheless paved the way for [^177^Lu]Lu-PSMA-617 treatment in patients with salivary gland carcinoma as a last resort. However, disease stabilization was shorter than that in the Phase III VISION study on prostate cancer with a median progression-free survival from 3.4 to 8.7 months. One of the explanations for these differences in results lies in both the lesser tracer uptake in tumors (median SUVmax of 8.2 in this study versus 13.3 in the VISION trial) and a lower number of cycles per patient (only two patients completed all four cycles in this study versus a median of five cycles per patient in the VISIOSN trial) [16]. Another recently published study included 28 salivary gland carcinoma metastatic patients and analyzed the diagnostic performance of [^68^Ga]Ga-PSMA-11 PET-CT and dosimetry, efficacy and safety of the [^177^Lu]Lu-PSMA-617 treatment in salivary gland malignancies [24]. This retrospective study analyzed [^68^Ga]Ga-PSMA-11 PET CT by two masked nuclear medicine physicians in joint consensus sessions and an additional board-certified radiologist for the CT reading session. All tumor images were classified as local tumor, regional lymph nodes, non-regional lymph nodes, lung, bone, and other regions. In each region, the tumors were classified into no evidence of disease, unifocal disease (one lesion), oligometastatic disease (two to five lesions), multifocal disease (six to ten lesions) and disseminated disease (more than ten lesions). The PSMA expression score was assessed visually for each patient in accordance with the PROMISE criteria which use blood pool, liver and parotid gland uptake as cut-off for low, intermediate and high PSMA expression on [^68^Ga]Ga-PSMA-11 PET CT [28]. The RLT consisted in 6.8 ± 1.4GBq of [^177^Lu]Lu-PSMA-617 administration every 6 weeks, and progression-free survival was studied according to RECIST1.1 [26]. Post-therapeutic dosimetry of tumor lesion and kidneys was calculated on post-therapeutic images with Single Photon Emission Computed Tomography (SPECT CT) with four time points according to the OLINDA/MIRD methods [29]. Twenty-four ACC, two adenocarcinoma, one acinic cell carcinoma and one sebaceous carcinoma patients were included. Results showed a superiority of the [^68^Ga]Ga-PSMA-11 PET CT compared with the CT scan with additional metastatic lesion in 11 patients with upstaging in 9 patients. The [^68^Ga]Ga-PSMA-11 PET CT detected more lesions compared to the CT scan in terms of local tumor (4%), regional lymph nodes (14%), distant lymph nodes (7%) and bone metastases (14%). The PSMA expression score according to the PROMISE criteria was two in 15 patients, three in 6 patients and less than two in 6 patients [28]. SUVmax was significantly higher for patients with a primary tumor in minor salivary glands (12.8 vs. 9) and in metastatic bone metastases (14.2 versus 6.4). The multidisciplinary tumor board validated the RLT option in each case if bone marrow and kidney function were adequate and for intermediate PSMA level expression in five patients. All these five patients had a tumor SUVmax superior to liver uptake, and a personalized dosimetry after each cycle of treatment was performed as previously described. The tumor dose and kidney dose were detailed for each patient on two target lesions. The five patients followed a total of 11 cycles of RLT (1 to 6 cycles). The tumor doses varied between 0.06 and 0.68 Gy/GBq and the kidney dose varied between 0.32 and 0.39 Gy/GBq among the five patients. One patient stopped cycles of treatment for insufficient tumor doses. No severe adverse events were noted and tumor stabilization over more than 1 year was observed for only one patient. Despite the small cohort and the number of patients treated, this retrospective study was able to identify patients who benefitted from treatment by RLT with a good tolerance and a tumor response. It is important to note than in this study, although high PSMA uptake was noted both on [^68^Ga]Ga-PSMA-11 PET CT and post-therapeutic images, retention times and tumor doses were lower and shorter than in prostate cancer, probably explaining the reduced effectiveness of the treatment compared to the VISION trial. In this study, only one patient of the five treated had a tumor stabilization over more than one year.

### 2.3. First Compassionate Use of [^177^Lu]Lu-PSMA-617 in Salivary Gland Cancer in France

It is in this context that we requested authorization from the French national drug agency called “Agence Nationale de Sécurité du Médicament” (ANSM) and from NOVARTIS for the compassionate use of [^177^Lu]Lu-PSMA-617 in a 56-year-old male patient treated for progressive metastatic salivary carcinoma. Indeed, this patient was initially diagnosed in 2018 with salivary duct carcinoma revealed by a cervical adenopathy with suspicion of a right submaxillary lesion and a single costal involvement. Initial surgical treatment consisted in bilateral cervical lymph node dissection and a rib metastatic resection.

Histopathologic findings suggested a salivary duct carcinoma without HER2 amplification but with overexpression of the androgen receptor. Unfortunately, he soon developed a new bone lesion in the pelvis and was administered carboplatin and taxotere chemotherapy as a first-line therapy. Evaluation after three courses showed a residual bone lesion for which a second-line treatment by immunotherapy (nivolumab) was prescribed. On the other hand, complementary local treatment by external radiation of some residual bone lesions was performed on the left iliac wing, left sacroiliac and the fifth thoracic vertebra. The immunotherapy was continued for about 2 years with good tolerance and no serious side effects. After an initial phase of stabilization of the disease, the therapeutic evaluation after 20 cycles showed an essentially bony progression with persistent pain, particularly in the pelvis. Since the primary tumor showed overexpression of androgen receptors, a complete androgenic blockade (anti-androgens and LHRH agonists (castration)) was offered to the patient as third-line treatment. The first assessment after 3 months of treatment with hormone therapy showed a stabilization of the disease correlated to a clinical improvement with a decrease in bone pain. However, six months later, the bone pain reappeared with a tendency for the bone metastases to progress. Unfortunately, we especially observed a life-threatening lesion on the second cervical vertebra (C2) requiring further radiation. Owing to previous radiation for a history of undifferentiated carcinoma of nasopharyngeal type (UCNT) several years prior, the therapeutic options to control the disease on C2 were limited, leading to combined treatment with a partial surgical resection and underdosed radiation.

It was in this context of the progressive continuation of the disease without systemic treatment possible as fourth-line therapy that treatment with [^177^Lu]Lu-PSMA-617 was envisaged. This therapeutic option was recommended by the IUCT-Oncopole multidisciplinary tumor board to the extent that no other therapeutic treatment was available for a young patient in good general condition.

In this perspective, first, it was necessary to evaluate the uptake of the disease on 68 [^68^Ga]Ga-PSMA-11 PET-CT (as previously explained in the context of theranostics) to ensure that the rate of uptake would be sufficient to treat the patient by RLT. [^68^Ga]Ga-PSMA-11 PET-CT revealed significant uptake in all known secondary lesions (mainly bone metastases) with a SUVmax of 10 for the hottest lesion localized in the left humeral head, and a SUVmax of 3 in the least intense one on the fifth thoracic vertebra (T5). The vast majority of lesions was more intense than in the liver, which presented a SUVmax of seven, which made it possible to consider treatment by RLT according to the PROMISE criteria, recently used as a reference for therapy by [^177^Lu]Lu-PSMA-617 [28]. Indeed, three quarters of the lesions had a level of expression of PSMA according to these criteria greater than or equal to two (Table 3)

As it is increasingly recommended now in RLT, we also performed a [^18^F]FDG (18F fluorodeoxyglucose) PET-CT. Indeed, [^18^F]FDG is the reference tracer to characterize tumor aggressiveness in almost all types of tumors, including well-differentiated tumors that may show an increased metastatic potential with dedifferentiation during their natural history (e.g., thyroid or prostate cancer). A SUVmax of 11.5 on this patient was localized on the lesion of the left humeral head and was the most intense one, and a SUVmax of 4.5 was identified on the right iliac wing and corresponded to the least intense one. In this context, compassionate use of [^177^Lu]Lu-PSMA-617 has been approved by Novartis and authorized by the ANSM. The patient consented to be treated after being informed of the benefit–risk ratio.

The first treatment was administered in the spring of 2022 with good clinical tolerance. The patient was treated every six weeks with a first evaluation after four courses (cumulative activity of 24.3 GBq of [^177^Lu]Lu-PSMA-617). Clinically, the symptomatology improved with a decrease in self-assessed pelvic bone pain from five to one after one cycle and a decrease in asthenia. Toxicity remained low with grade 1 anemia (pre-existing to [^177^Lu]Lu-PSMA-617 treatment and related to previous treatments), grade 3 lymphopenia (without clinical repercussions and usually observed with such treatments) and grade 2 dry mouth (a frequently reported adverse effect in relation with the physiologic parotid gland uptake of PSMA) according to the National Cancer Institute Common terminology criteria for adverse events (CTCAE) version 5.0 [27]. We performed an imaging assessment after four courses of RLT. The imaging evaluation consisted in a functional one by [^18^F]FDG PET CT and a morphological one by CT scan. Imaging showed stable disease according to PERCIST criteria with no significant variation in SUVmax on the hottest lesion on 18FDG PET-CT before and after RLT [25]. However, the C2 uptake was more widespread, and a complementary cervical MRI confirmed the instability of the lesion. Due to the location of this metastasis, we decided to propose a neurosurgical stabilization that occurred after four cycles. Similarly, the CT scan showed a stable disease according to the RECIST criteria without no new lesions [26] (Figure 1).

In parallel, dosimetry was performed. After each cycle, we determined four time points and data from SPECT CT contouring some target metastases and healthy organ (kidney, medulla by the average of lumbar vertebrae 2, 3 and 4) using a dosimetry software (PlanetDose). After four courses, the cumulative dose was 942.3 mGy for the whole body, 9.8 Gy (0.39 Gy/GBq) for the left kidney, 10.6 Gy (0.43 Gy/GBq) for the right kidney, and 0.3 Gy (0.012 Gy/GBq) for the medulla. After three courses of treatment, the cumulative dose in the C2 lesion and the humerus lesion was 0.6 Gy (0.37 Gy/GBq) and 0.3 Gy (0.04 Gy/GBq), respectively. Considering the disease stabilization after four cycles, we obtained the authorization to follow RLT. The patient is currently under treatment with two additional courses in order to stabilize the disease.

## 3. Discussion

Therapy for salivary gland cancer is challenging at a metastatic stage with few systemic options available, especially if targeted therapy or hormonal treatment are not feasible. All systemic treatments have several side effects that can often lead to discontinuation of the drug or a decrease in dosage and therefore effectiveness. In addition, it is difficult to control the disease for a long time, so other alternatives are needed. The first results of [^177^Lu]Lu-PSMA-617 treatment in castration-resistant prostate cancer are very encouraging. Indeed, toxicities are low with mainly only grade 1 and 2 side effects. In the Phase III VISION trial, there is a modest but real gain in overall survival (15.3 months versus 11.3 months under the best supportive care) and progression-free survival (8.7 months versus 3.4 months under the best supportive care) in last-line castration-resistant prostate cancer patients compared to that of the best supportive care [16].

Reports on the treatment of salivary gland cancers with [^177^Lu]Lu-PSMA-617 are few due to the rarity of the disease. However, they suggest that RLT is well-tolerated in this population, with interesting and encouraging results in metastatic disease after one or more lines of systemic treatment. However, as suggested in the recent study of Civan et al. [24], the rate of [^177^Lu]Lu-PSMA-617 uptake is lower and retention times of PSMA uptake are probably shorter than those in castration-resistant prostate cancer, so its efficacy is also lower due to a lower tumor dose received. As a result, large-scale dosimetric studies would be needed to evaluate the optimal doses for the effective uptake of [^177^Lu]Lu-PSMA-617 first in patients treated for prostate cancers. These data would allow determining the optimal anti-tumor dose required in prostate cancer. In future studies, the dosimetry could preferentially be applied to salivary gland carcinoma patients treated by RLT.

We could consider several options to improve the response of salivary gland carcinoma to RLT. The first one acts on the received dose, whatever the residence time, i.e., the increase in the administered activity would allow a higher activity within each tumor lesion. All this has to be balanced against the potential side effects induced by RLT, particularly radiation-induced cancers. The second option could be the use of radio-sensitizers or the association with other treatment such as chemotherapy during the courses of [^177^Lu]Lu-PSMA-617. Again, the cumulative toxicity of treatments must be taken into account in this option. The third track could be the use of alpha emitters combined with PSMA. This alpha therapy induces a deposit of energy much more important and on a much shorter range than with the ^177^Lu, allowing both to irradiate more tumors and to better preserve healthy tissue (Table 4).

[^225^Ac]Ac-PSMA-617 was developed with the aim of offering a second RLT to patients who do not respond to [^177^Lu]Lu-PSMA-617. [^225^Ac]Ac-PSMA-617 is already used in some countries and trials but with controversial effects regarding toxicity on salivary gland in particular [31,32]. Indeed, the first review from Lunger et al. [32] reports that [^225^Ac]Ac-PSMA-617 could be a promising alternative for patients developing progression under [^177^Lu]Lu-PSMA-617 but would induce mostly permanent xerostomia. Another more recent review [17] shows a low morbidity profile of [^225^Ac]Ac-PSMA-617 and a loss of efficacy in patients pre-treated with [^177^Lu]Lu-PSMA-617. Further randomized controlled trials are required to clarify these data and position [^225^Ac]Ac-PSMA-617 in relation to [^177^Lu]Lu-PSMA-617. New data concerning [^225^Ac]Ac-PSMA-617 therapy could make it possible to consider this treatment in salivary gland cancers and better assess the benefit–risk balance in this type of rare tumor.

A last option to improve local radiation and decrease toxicity could be locoregional application as used with the Selective Internal Radiation Therapy (SIRT) for hepatic primitive or secondary tumor treatment. This technique allows to bring by the arterial way as close as possible to the tumor radioactive beads, allowing high irradiation near the tumor while sparing a large part of the healthy organs (adjacent healthy liver in particular). In our cases, however, the disease presents distant metastases not only limited to an organ whose vascularization would be accessible. Therefore, this option does not appear to be applicable to metastatic salivary gland carcinoma.

These associations and other options for [^177^Lu]Lu-PSMA-617 might prove efficient and warrant further investigation. Other interesting ways to improve the efficacy of [^177^Lu]Lu-PSMA-617 in salivary gland carcinoma could also exist.

## 4. Conclusions

The scarce published data from the few patients treated with [^177^Lu]Lu-PSMA-617 and the patient treated on a compassionate basis in France are in favor of a trend towards stabilization of the disease. These results are encouraging in metastatic patients who have no further therapeutic recourse and would prompt the proposal of randomized multicenter studies including more patients. Indeed, [^177^Lu]Lu-PSMA-617 appears to be a promising therapeutic option as last-line treatment for metastatic salivary gland carcinoma, provided its uptake on [^68^Ga]Ga-PSMA-11 PET-CT is sufficient. Although treatment with [^177^Lu]Lu-PSMA-617 is currently problematic in France for administrative reasons, the prospect of treating patients with rare cancers at a metastatic stage such as salivary carcinomas opens up new therapeutic options and management strategies.

## 5. Future Research

There is now need for a Phase III study with dosimetric data comparing the efficacy of [^177^Lu]Lu-PSMA-617 to the best available treatment as last-line treatment for metastatic salivary gland carcinoma in order to determine the optimal tumor dose for higher efficacy. This type of study remains difficult to carry out because of the rarity of this type of tumor and the small number of centers with the possibility of using [^177^Lu]Lu-PSMA-617. It would be interesting to evaluate the association of [^177^Lu]Lu-PSMA-617 and radio-sensitizers such as chemotherapy, the alpha PSMA therapy or the increase in [^177^Lu]Lu-PSMA-617 activity to improve the efficacy of RLT in salivary gland carcinomas with moderate uptake of PSMA. In the same way, it would be interesting to consider analyzing the combination of RLT with other systemic treatments already used in these types of cancers, notably hormonal therapies and targeted therapy. This type of large-scale study requires a lot of time and the participation of many centers to obtain reliable results and lead to the use of [^177^Lu]Lu-PSMA-617 in these indications. Consideration should also be given to competitive access to RLT for rare cancers to allow patients without therapeutic recourse to have access to this treatment. Such access has to be controlled and monitored by the competent authorities to ensure that there is a benefit–risk balance in favor of the treatment.

## Figures and Tables

**Figure 1 pharmaceuticals-16-00754-f001:**
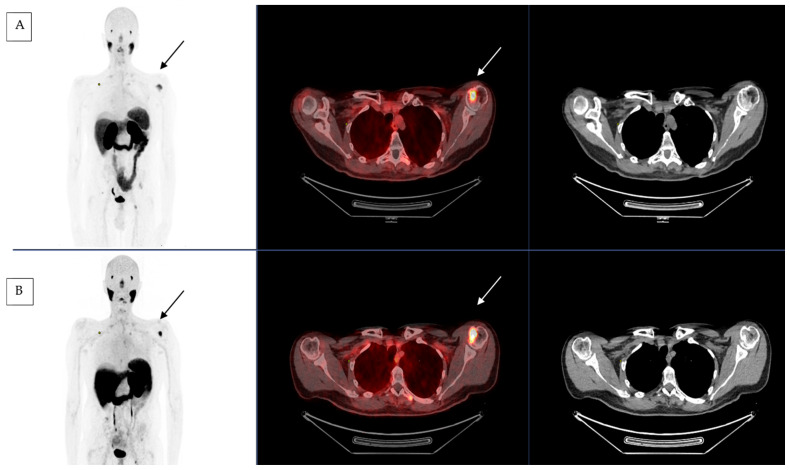
[^68^Ga]Ga-PSMA-11 PET-CT before (**A**) and after (**B**) 4 cycles of RLT by [^177^Lu]Lu-PSMA-617 with left humeral head lesion (black and white arrow).

**Table 1 pharmaceuticals-16-00754-t001:** World Health Organization classification of tumors, pathology and genetics of head and neck tumors for malignant salivary gland tumors (2005) and expression of androgen receptors, HER 2 and PSMA expression [2,3].

Malignant Epithelial Tumors	Androgen Receptor Expression	HER 2 Expression (%)	PSMA Expression (%)
Acinic cell carcinoma		16.2	94
Mucoepidermoid carcinoma		12.9	
Adenoid cystic carcinoma		17.2	
Polymorphous low-grade adenocarcinoma		4.0	
Epithelial–myoepithelial carcinoma		5.2	
Clear cell carcinoma, not otherwise specified			
Basal cell adenocarcinoma		4.7	
Sebaceous carcinoma			
Sebaceous lymphadenocarcinoma			
Cystadenocarcinoma		3.3	
Low-grade cribriform cystadenocarcinoma			
Mucinous adenocarcinoma			
Oncocytic carcinoma		4.6	
Salivary duct carcinoma	Often	44.9	62.5
Adenocarcinoma, not otherwise specified	Often	22.3	
Myoepithelial carcinoma		5.1	
Carcinoma ex pleomorphic adenoma		33.7	
Carcinosarcoma			
Metastasizing pleomorphic adenoma			
Squamous cell carcinoma		21.4	
Small cell carcinoma			
Large cell carcinoma			
Lymphoepithelial carcinoma			
Sialoblastoma			

**Table 2 pharmaceuticals-16-00754-t002:** Studies about [^177^Lu]Lu-PSMA-617 in salivary gland carcinoma. IHC: No: number; ImmunoHistoChemistry; PR: Partial Response; SD: Stable Disease; PD: Progressive Disease; NA: Not Applicable.

Study Author (Year of Publication)	Type of Study	No. of Patients	Tumor Type	PSMA Expression (% by IHC)	[^68^Ga]Ga-PSMA-11 PET CT	SUV Max	[^177^Lu]Lu-PSMA-617	No of Cycles	Response to Treatment	Toxicities	Dosimetry
**Has Simsek et al. 2019 [22]**	Case report	1	ACC	No data	Yes (1)	No data	yes	1	Bone pain decrease	None	No data
**Klein Nulent et al. 2021 [23]**	Retrospective	6	ACC (4) Adenoma NOS (1) Acinic cell carcinoma (1)	5 to 95	Yes (6)	3.5 to 10.2	Yes	1 to 4	PR (1) SD (1) PD (3) NA (1)	Grade 1–2	No data
**Civan et al. 2023** **[24]**	Retrospective	5 (from 28 for PET imaging)	ACC (4) Acinic cell carcinoma (1)	No data	Yes (28)	10.7 (average)	Yes (5 from 28)	1 to 6	SD (2) PD (1) NA (2)	SD (2) PD (1) NA (2) None	Yes (olinda/MIRD)

**Table 3 pharmaceuticals-16-00754-t003:** miPSMA expression score according to the PROMISE criteria [28].

Score	Reported PSMA Expression	Uptake
0	No	Below blood pool
1	Low	Equal to or above blood pool and lower than liver *
2	Intermediate	Equal to or above liver and lower than parotid gland
3	High	Equal to or above parotid gland

* For PSMA ligands with liver-dominant excretion (e.g., [^18^F]F-PSMA-1007) spleen is recommended as reference organ instead of liver.

**Table 4 pharmaceuticals-16-00754-t004:** Characteristics of alpha and beta particle radiation [30].

	Alpha Particle Radiation	Beta Particle Radiation
Energy	5–9 meV	50–2300 keV
Range	40–100 um	0.05–12 mm
Linear energy transfer	80 keV/mm	0.2 keV/mm
Mass	4 amu	1/2000 amu
Charge	3.1 × 10^−19^ C	1.6 × 10^−19^ C
Penetrate power	10^1^ mm Al, 3–8 cm air	5 mm Al, 1 mm lead

## Data Availability

The data presented in this study are available on request from the corresponding author. The data are not publicly available due to ethical restriction.

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
