# Peer review of "RadioLigand Therapy with [^177^Lu]Lu-PSMA-617 for Salivary Gland Cancers: Literature Review and First Compassionate Use in France"

_pharmaceuticals, 2023, doi:10.3390/ph16050754_

Round 1
Reviewer 1 Report
It is a nice review but should be implemented and more focused on salivary gland tumors create with 177Lu-PSMA.
Indeed, there are very few reports on 177Lu-PSMA treatment of salivary gland tumors and authors should concentrate more on these findings and make one or more tables to highlight pros and cons, population, outcome, doses, etc.
In figure 1 the arrows do not point at the lesions.
Author Response
Please find a revised version of th manuscript entitled “RadioLigand Therapy with [177Lu]Lu-PSMA-617 for salivary gland cancers: literature review and first compassionate use in France ”.
As suggested, the manuscript was revised and we answered point by point to the critiques made by the reviewers. We hope the improvement of the manuscript will make it acceptable for publication.
Thank you for your time and consideration.
Sincerely
For all authors
Marie Terroir MD, (first author and corresponding author)
it is a nice review but should be implemented and more focused on salivary gland tumors create with 177Lu-PSMA.
This paper is only focused on salivary gland carcinoma treated by 177Lu-PSMA
Indeed, there are very few reports on 177Lu-PSMA treatment of salivary gland tumors and authors should concentrate more on these findings and make one or more tables to highlight pros and cons, population, outcome, doses, etc.
We have added a summary table (table 1) in the main text.
Table 2. Studies about 177Lu-PSMA617 in salivary gland carcinoma (please see the attachment)
In figure 1 the arrows do not point at the lesions
Thank you, we apologize we have corrected the mistake

Reviewer 2 Report
This paper is more likely a single-case report on off-label use of 177Lu-PSMA-617 compassionate use supported by a literature and current status review instead of being qualified for a full review paper.
In the introduction part I am missing an overview on PSMA expression in salivary gland cancers - that would certainly be of general interest.
Figure 1 – please correct the arrows position.
Page 3, section 2 – This section should be subdivided in more specific parts. Definitely it would be very helpful to separate the compassionate use conditions with more detailed description of the treatment protocol. Please reorganize this section.
In discussion part, I would suggest to discuss in more detail the possibilities of further tumor uptake improvement and/or kidney dose lowering. Would locoregional application of 177Lu-PSMA-617 be an option?
In my opinion, this paper is more likely a single-case report on compassionate, off-label, use of 177Lu-PSMA-617 for the treatment of salivary gland cancer supported by literature review, instead of being qualified to a full review paper.
In Table 1 I am missing some data on PSMA expression that would be of major interest to the readers instead of HER2 receptors expression.
Conclusion should contain more detailed assessment of the treatment outcome instead of rather general statements about 177Lu-PSMA-617 use.
Author Response
Please find a revised version of th manuscript entitled “RadioLigand Therapy with [177Lu]Lu-PSMA-617 for salivary gland cancers: literature review and first compassionate use in France ”.
As suggested, the manuscript was revised and we answered point by point to the critiques made by the reviewers. We hope the improvement of the manuscript will make it acceptable for publication.
Thank you for your time and consideration.
Sincerely
For all authors
Marie Terroir MD, (first author and corresponding author)
This paper is more likely a single-case report on off-label use of 177Lu-PSMA-617 compassionate use supported by a literature and current status review instead of being qualified for a full review paper.
It is indeed a review of the only 3 publications on salivary gland carcinomas treated by 177Lu PSMA-617 and on a clinical case of a patient treated by 177Lu PSMA-617 in France on a compassionate basis.
In the introduction part I am missing an overview on PSMA expression in salivary gland cancers - that would certainly be of general interest.
We have added 3 references about PSMA expression in ACC and salivary duct carcinoma. We added the proportion of expression in Table 1 and comments l 93-97.
Figure 1 – please correct the arrows position
Thank you, we apologize and have corrected the mistake
Page 3, section 2 – This section should be subdivided in more specific parts. Definitely it would be very helpful to separate the compassionate use conditions with more detailed description of the treatment protocol. Please reorganize this section.
We have devided section 2 in :
- A : 177Lu PSMA-617 use in prostate cancer
- B : 177Lu PSMA-617 use in salivary gland carcinoma
- C : First compassionate used of [177Lu]Lu-PSMA-617 in salivary gland cancer in France
In discussion part, I would suggest to discuss in more detail the possibilities of further tumor uptake improvement and/or kidney dose lowering. Would locoregional application of 177Lu-PSMA-617 be an option?
We have added discussion about locoregional option in l 338-345
In my opinion, this paper is more likely a single-case report on compassionate, off-label, use of 177Lu-PSMA-617 for the treatment of salivary gland cancer supported by literature review, instead of being qualified to a full review paper.
It is indeed a review of the only 3 publications on salivary gland carcinomas treated by 177Lu PSMA-617 and on a clinical case of a patient treated by 177Lu PSMA-617 in France on a compassionate basis
In Table 1 I am missing some data on PSMA expression that would be of major interest to the readers instead of HER2 receptors expression.
We have added the proportion of expression of PSMA in Table 1
Conclusion should contain more detailed assessment of the treatment outcome instead of rather general statements about 177Lu-PSMA-617 use.
Thank you for your suggestion. We have summarized the observed results of the treatment on the few patients treated.

Reviewer 3 Report
Marie Terroir et al. conducted a thorough evaluation of salivary gland tumors and used the PSMA targeting drug 177Lu177-PSMA as an alternative therapeutic option for metastatic salivary gland tumors. It is also encouraging to learn of the first compassionate use in France.
The paper under consideration is a well-written and thorough narrative assessment of the use of 177Lu-PSMA-based Theranostics for salivary gland treatment. I congratulate the authors on their outstanding article. I suggest a minor correction.
1) I recommend discussing salivary gland toxicity with mCRPC patients treated with 177Lu-PSMA. Here are a few papers for your consideration.
• Toxicity of PSMA-Targeted Radioligand Therapy with 177Lu-PSMA and Combined 225Ac- and 177Lu-Labeled PSMA Ligands (TANDEM-PRLT) in Advanced Prostate Cancer: A Single-Center Systematic Investigation, Diagnostics 2022, 12(8), 1926
• Salivary gland function after 177Lu-PSMA radioligand therapy: Current imaging and treatment approaches, Translational Oncology, 2022, 21, 101445.
2) There is substantial preclinical and clinical data on mRPC with 225Ac-PSMA in the literature. As a result, I advocate including more discussion of the pros and cons of employing the same drug in the salivary gland.
3) I propose that more figures from the cited literature be included in the introduction and discussion parts.
4) I propose including the time-period gathering literature for the review.
5) Please correct a few spelling errors detected in 2) pertinent portions (lines 97, 111, 128, 177, 190, 212).
Moderate English corrections are required.
Author Response
Please find a revised version of th manuscript entitled “RadioLigand Therapy with [177Lu]Lu-PSMA-617 for salivary gland cancers: literature review and first compassionate use in France ”.
As suggested, the manuscript was revised and we answered point by point to the critiques made by the reviewers. We hope the improvement of the manuscript will make it acceptable for publication.
Thank you for your time and consideration.
Sincerely
For all authors
Marie Terroir MD, (first author and corresponding author)
Marie Terroir et al. conducted a thorough evaluation of salivary gland tumors and used the PSMA targeting drug 177Lu177-PSMA as an alternative therapeutic option for metastatic salivary gland tumors. It is also encouraging to learn of the first compassionate use in France.
The paper under consideration is a well-written and thorough narrative assessment of the use of 177Lu-PSMA-based Theranostics for salivary gland treatment. I congratulate the authors on their outstanding article. I suggest a minor correction.
We thank you for your comments and your encouragement
1) I recommend discussing salivary gland toxicity with mCRPC patients treated with 177Lu-PSMA. Here are a few papers for your consideration.
- Toxicity of PSMA-Targeted Radioligand Therapy with 177Lu-PSMA and Combined 225Ac- and 177Lu-Labeled PSMA Ligands (TANDEM-PRLT) in Advanced Prostate Cancer: A Single-Center Systematic Investigation, Diagnostics 2022, 12(8), 1926
- Salivary gland function after 177Lu-PSMA radioligand therapy: Current imaging and treatment approaches, Translational Oncology, 2022, 21, 101445.
Thank you for your suggestion. We have inserted some information on salivary toxicity (l 118-123) and references
2) There is substantial preclinical and clinical data on mRPC with 225Ac-PSMA in the literature. As a result, I advocate including more discussion of the pros and cons of employing the same drug in the salivary gland.
As suggested, we have added some discussion about 225Ac PSMA (l 350-362)
3) I propose that more figures from the cited literature be included in the introduction and discussion parts.
We have found it difficult to introduce figures from the literature but we added a summary table (table 2) of the 3 publications concerning 177Lu-PSMA and salivary gland carcinomas (please see the attachment)
4) I propose including the time-period gathering literature for the review.
We have added some details about the review method (l 124-129) and the time-period
5) Please correct a few spelling errors detected in 2) pertinent portions (lines 97, 111, 128, 177, 190, 212).
Thank you, we have corrected the spelling errors in the text.
Moderate English corrections are required

Reviewer 4 Report
In the paper “RadioLigand Therapy with 177Lu PSMA for salivary gland cancers: literature review and first compassionate use in France”, Dr. Terroir et al. gives a review on the literature of the literature on the subject, including the theranostic pair of [68Ga]Ga-PSMA-11 for diagnostics and [177Lu]Lu-PSMA-617 for therapy, and combined with a description of their own data from France, concluding that this is a promising treatment that should proceed to phase III trial.
This is an interesting and relevant subject, but the presentation should be strengthened, with a more clear distinction between literature review and reporting of own data.
MAJOR
1. A standard reporting form has sections of Introduction, Material & Methods, Results, Discussion, Conclusion. This standard is not necessarily the best for all papers, especially reviews may need a different form. Still, the present section 2 of the paper becomes too unfocused, with a mix of review, methods and (own or reviewed?) results combined in a single section. The authors are advised to split this section into more focused sections. It would also give a better overview if a table was added, summarizing the reviewed studies and how many patients were part of each study.
MINOR
2. Generally on radiopharmaceutical notation: A widely accepted consensus paper with guidelines for radiochemistry nomenclature was published in 2017. According to the guideline, radiopharmaceuticals should be written with the full molecule (including the atom used for labelling), with the specific isotope in square parentheses right to the left of this. E.g. [177Lu]Lu-PSMA-617 (the full molecule is Lu-PSMA-617, lutetium is not part of PSMA-617), [68Ga]Ga-PSMA-11, and [18F]FDG (FDG is the full molecule, which includes the radioactive fluorine atom). Please note that the numbers in 177Lu etc. should be in superscript. The guideline referred to is summarized in this (widely published) open letter: Coenen HH et al. Open letter to journal editors on: International consensus radiochemistry nomenclature guidelines. EJNMMI Radiopharmacy and Chemistry (2019); 4(1). https://doi.org/10.1186/s41181-018-0047-y
3. Lines 273-280: This paragraphs opens by stating that “dosimetry was performed”, but gives not description HOW dosimetry was performed. Please describe, at least briefly, how dosimetry was performed.
4. Line 93: Section 2 is titled “Relevant section”, which is not quite clear. Perhaps a more focussed section title could be used?
5. Line 94: Even though the abbreviation RLT (radioligand therapy) was defined in the abstract, please define it (again) here at this first mention in the main text of the article.
6. Title, line 107 and line 133: The variant of PSMA is not specified. Presumably it is PSMA-617 in the title and line 133, and PSMA-11 in line 107?
7. Line 240: The tracer “18FPSMA1007” appears to be [18F]F-PSMA-1007, correct?
8. Line 332: Is it on purpose that no number variant of PSMA is given at this mention of PSMA RLT (“PSMARLT”)? Without a number, it would indicate (lutetium-labelled) PSMA RLT in general, with a number, it would be more specific, referring to [177Lu]Lu-PSMA-617 RLT as it is the topic elsewhere in this review.
9. Line 365: Reference 2 appears incomplete. What publication from REFCOR is referred to? Please give title of the publication, optimally supplied by a link to the publication.
TYPOGRAPHICAL ISSUES:
Line 19: ”1177” should be ”177” (but see also general comment on radiopharmaceutical notation).
Line 97: ant -> and
Line 111: opase -> phase
Line 111: theranoscis -> theranostics (missing t)
Line 128: T the -> The
In a number of places, double spaces appear. The authors are advised to perform a search for double spaces and check if not most of these were meant to be just single spaces.
Author Response
Please find a revised version of th manuscript entitled “RadioLigand Therapy with [177Lu]Lu-PSMA-617 for salivary gland cancers: literature review and first compassionate use in France ”.
As suggested, the manuscript was revised and we answered point by point to the critiques made by the reviewers. We hope the improvement of the manuscript will make it acceptable for publication.
Thank you for your time and consideration.
Sincerely
For all authors
Marie Terroir MD, (first author and corresponding author)
MAJOR
- A standard reporting form has sections of Introduction, Material & Methods, Results, Discussion, Conclusion. This standard is not necessarily the best for all papers, especially reviews may need a different form. Still, the present section 2 of the paper becomes too unfocused, with a mix of review, methods and (own or reviewed?) results combined in a single section. The authors are advised to split this section into more focused sections. It would also give a better overview if a table was added, summarizing the reviewed studies and how many patients were part of each study.
We have added some details about the review method (l 124-129) and Table 2 to summarize the three relevant studies. (please see the attachment)
MINOR
- Generally on radiopharmaceutical notation: A widely accepted consensus paper with guidelines for radiochemistry nomenclature was published in 2017. According to the guideline, radiopharmaceuticals should be written with the full molecule (including the atom used for labelling), with the specific isotope in square parentheses right to the left of this. E.g. [177Lu]Lu-PSMA-617 (the full molecule is Lu-PSMA-617, lutetium is not part of PSMA-617), [68Ga]Ga-PSMA-11, and [18F]FDG (FDG is the full molecule, which includes the radioactive fluorine atom). Please note that the numbers in 177Lu etc. should be in superscript. The guideline referred to is summarized in this (widely published) open letter: Coenen HH et al. Open letter to journal editors on: International consensus radiochemistry nomenclature guidelines. EJNMMI Radiopharmacy and Chemistry (2019); 4(1). https://doi.org/10.1186/s41181-018-0047-y
Thank you, we have modified all radiopharmaceutical notation.
- Lines 273-280: This paragraphs opens by stating that “dosimetry was performed”, but gives not description HOW dosimetry was performed. Please describe, at least briefly, how dosimetry was performed.
We have proposed to add more details about dosimetry in line 307-309
- Line 93: Section 2 is titled “Relevant section”, which is not quite clear. Perhaps a more focussed section title could be used?
Unfortunately these are sections imposed by the newspaper
- Line 94: Even though the abbreviation RLT (radioligand therapy) was defined in the abstract, please define it (again) here at this first mention in the main text of the article.
As suggested, we have defined RLT again at the first mention in the main text
- Title, line 107 and line 133: The variant of PSMA is not specified. Presumably it is PSMA-617 in the title and line 133, and PSMA-11 in line 107?
Yes, you are right. All treatment described in this review was PSMA-617. We precised PSMA-617 in title and in line 133. We have corrected PSMA-11 in line 107.
- Line 240: The tracer “18FPSMA1007” appears to be [18F]F-PSMA-1007, correct?
Yes, we have corrected “18FPSMA1007” by [18F]F-PSMA-1007
- Line 332: Is it on purpose that no number variant of PSMA is given at this mention of PSMA RLT (“PSMARLT”)? Without a number, it would indicate (lutetium-labelled) PSMA RLT in general, with a number, it would be more specific, referring to [177Lu]Lu-PSMA-617 RLT as it is the topic elsewhere in this review.
We agree. We have modified PSMA RLT by [177Lu]Lu-PSMA-617
- Line 365: Reference 2 appears incomplete. What publication from REFCOR is referred to? Please give title of the publication, optimally supplied by a link to the publication
it is a consensus of French experts available on : http://refcor.org/files/116/recommandations/refcor_glandes_salivaires.pdf
TYPOGRAPHICAL ISSUES:
Line 19: ”1177” should be ”177” (but see also general comment on radiopharmaceutical notation).
Line 97: ant -> and
Line 111: opase -> phase
Line 111: theranoscis -> theranostics (missing t)
Line 128: T the -> The
Thank you, we have corrected the spelling errors in the text.
In a number of places, double spaces appear. The authors are advised to perform a search for double spaces and check if not most of these were meant to be just single spaces
Thank you, we have removed double spaces in the text.

Round 2
Reviewer 1 Report
All corrections have been made as requested.
Author Response
Thank you for your suggestions and your time
Reviewer 4 Report
This revised version of the paper on [177Lu]Lu-PSMA-617 for salivary gland cancers is overal satisfactorily improved.
A number of very minor and typographical issues would be in the common interest (authors, editor, reviewers) of providing the readers the most readable version of the authors’ work. These are specified below.
MINOR
Both Table 1 and Table 2 are split over two pages. Readability would be increased if table splitting was avoided. For Table 1 this could be achieved by moving the table one paragraph down. This would still present the table shortly after its first mention. Table 2 could perhaps be moved upward to be place between section 2.a and 2.b.
Table 2: The formatting of the table results in many words being split, even without a hyphen. Here follows some suggestions for reformatting (meant as suggestions for inspiration, not as dictate). First, it is suggested to use the full width to make the right margin of the table follow the right margin of the text. Second, it is suggested to rethink he distribution of the space among the columns. A starting point may be to make all columns the same width, and from there consider if some columns need more space than others. For instance, the first column with author names and references, probably need more space than columns giving information like “Yes (1)” (6th column) or “1-4” (9th column). Regarding Table 2, please also consider moving the table legend (“Table 2. Studies ...”) above the table, and place the solid line above the table headings, as was done in Tables 1 and 3.
Title of section 2: Based on own publication in MDPI journals, it is the impression of this reviewer that template title “Relevant section” is meant only as a template to be filled in with the text relevant for the present paper. It would be more descriptive with a section title like “Review of [177Lu]Lu-PSMA-617 in France”, or something else to the authors’ preference.
Lines 194 and 252: Is it correctly understood that “68” in both these places should be deleted? I.e., it is not about 68 scans, the 68 is a left-over from the reformatting of the tracer name?
Line 338: “a much shorter route”. While this wording is not wrong, it would be more specific to write “a much shorter range” (route -> range), in accordance with the term used in Table 3.
Lines 328-329: This reviewer is a bit uncertain about this sentence: “In a second time, the dosimetry could be applied to patient treated by RLT for salivary gland carcinoma.” Is the sentence about a second therapy for a specific (single) patient? Or is it a suggestion regarding future patients in general? If the meaning is general, it is suggested to rephrase, e.g.: “In future studies, ... could preferentially be applied to patients [plural] ...”
Line 330: The word “hypothesis” is most often used about a study hypothesis to be tested. In the present case, the intention seems to be more like “suggestions” or “ideas”? Furthermore, it is suggested to let the sentence “We could consider ...” begin a new paragraph rather than continue the previous paragraph.
Reference 4. It is suggested to include the report title and the link in the reference: Aegerter DP, Cosmidis DA. Recommandation pour la Pratique Clinique, G4 – Tumeurs malignes primitives des glandes salivaires. Bureau REFCOR. Available at: http://refcor.org/files/116/recommandations/refcor_glandes_salivaires.pdf
TYPOGRAPHICAL
Line 115: “dry mouse” -> “dry mouth”
Line 194 and a few other places: “PET CT” should be “PET-CT” (or “PET/CT”, in which case this term should be consistently used)
Line 203: “SUV max” -> “SUVmax” (no space)
Line 207: “gy” -> “Gy”
Line 307: If this headline is still on a single line after reformatting the table, please move it to the next page. A smart feature for this in Word is to right-click, choose paragraph formatting, click tab “line and page shifts” and here click “Keep together with next”. This automatically keeps headline together with the following text, without the need to manually add or remove extra blank lines.
Lines 338, 372 and 399: Please make 177 superscript.
Line 373: The repetition “177Lu-PSMA-617” should be deleted.
Line 399: Please check the radiopharmaceutical name.
See specific comments in the general comments.
Author Response
This revised version of the paper on [177Lu]Lu-PSMA-617 for salivary gland cancers is overal satisfactorily improved.
Thank you for your return and all yours suggestions.
A number of very minor and typographical issues would be in the common interest (authors, editor, reviewers) of providing the readers the most readable version of the authors’ work. These are specified below.
MINOR
Both Table 1 and Table 2 are split over two pages. Readability would be increased if table splitting was avoided. For Table 1 this could be achieved by moving the table one paragraph down. This would still present the table shortly after its first mention. Table 2 could perhaps be moved upward to be place between section 2.a and 2.b.
We have placed table 2 between section 2a and 2b
Table 2: The formatting of the table results in many words being split, even without a hyphen. Here follows some suggestions for reformatting (meant as suggestions for inspiration, not as dictate). First, it is suggested to use the full width to make the right margin of the table follow the right margin of the text. Second, it is suggested to rethink he distribution of the space among the columns. A starting point may be to make all columns the same width, and from there consider if some columns need more space than others. For instance, the first column with author names and references, probably need more space than columns giving information like “Yes (1)” (6th column) or “1-4” (9th column). Regarding Table 2, please also consider moving the table legend (“Table 2. Studies ...”) above the table, and place the solid line above the table headings, as was done in Tables 1 and 3.
We have corrected the formatting of the table 2
Title of section 2: Based on own publication in MDPI journals, it is the impression of this reviewer that template title “Relevant section” is meant only as a template to be filled in with the text relevant for the present paper. It would be more descriptive with a section title like “Review of [177Lu]Lu-PSMA-617 in France”, or something else to the authors’ preference.
Thank you for the clarification. We have modified by « use of [177Lu]Lu-PSMA-617 in France »
Lines 194 and 252: Is it correctly understood that “68” in both these places should be deleted? I.e., it is not about 68 scans, the 68 is a left-over from the reformatting of the tracer name?
Yes you are right We have corrected the mistake
Line 338: “a much shorter route”. While this wording is not wrong, it would be more specific to write “a much shorter range” (route -> range), in accordance with the term used in Table 3.
We have modified « route » by « range » line 338
Lines 328-329: This reviewer is a bit uncertain about this sentence: “In a second time, the dosimetry could be applied to patient treated by RLT for salivary gland carcinoma.” Is the sentence about a second therapy for a specific (single) patient? Or is it a suggestion regarding future patients in general? If the meaning is general, it is suggested to rephrase, e.g.: “In future studies, ... could preferentially be applied to patients [plural] ...”
You are right the sense is general. We have modified the sentence line 329 as suggested.
Line 330: The word “hypothesis” is most often used about a study hypothesis to be tested. In the present case, the intention seems to be more like “suggestions” or “ideas”? Furthermore, it is suggested to let the sentence “We could consider ...” begin a new paragraph rather than continue the previous paragraph.
We have changed « hypothesis » by « options » line 331 and have begun a new paragraph.
Reference 4. It is suggested to include the report title and the link in the reference: Aegerter DP, Cosmidis DA. Recommandation pour la Pratique Clinique, G4 – Tumeurs malignes primitives des glandes salivaires. Bureau REFCOR. Available at: http://refcor.org/files/116/recommandations/refcor_glandes_salivaires.pdf
Thank you for the suggestion. We have modified the reference
TYPOGRAPHICAL
Line 115: “dry mouse” -> “dry mouth”
Thank you. We have corrected the mistake
Line 194 and a few other places: “PET CT” should be “PET-CT” (or “PET/CT”, in which case this term should be consistently used)
Thank you. We have corrected the mistake
Line 203: “SUV max” -> “SUVmax” (no space)
Thank you. We have corrected the mistake
Line 207: “gy” -> “Gy”
Thank you. We have corrected the mistake
Line 307: If this headline is still on a single line after reformatting the table, please move it to the next page. A smart feature for this in Word is to right-click, choose paragraph formatting, click tab “line and page shifts” and here click “Keep together with next”. This automatically keeps headline together with the following text, without the need to manually add or remove extra blank lines.
Thank you. We have corrected the mistake
Lines 338, 372 and 399: Please make 177 superscript.
Thank you. We have corrected the mistake
Line 373: The repetition “177Lu-PSMA-617” should be deleted.
Thank you. We have corrected the mistake
Line 399: Please check the radiopharmaceutical name.
Thank you. We have corrected the mistake